# Frequency-selective perovskite photodetector for anti-interference optical communications

Liangliang Min[1], Haoxuan Sun [1]✉, Linqi Guo[1], Meng Wang[1], Fengren Cao[1], Jun Zhong[2] & Liang Li [1]✉

Free-space coupling, essential for various communication applications, often faces significant signal loss and interference from ambient light. Traditional methods rely on integrating complex optical and electronic systems, leading to bulkier and costlier communication equipment. Here, we show an asymmetric 2D–3D–2D perovskite structure device to achieve a frequency-selective photoresponse in a single device. By combining two electromotive forces of equal magnitude in the opposite directions, the device output is attenuated to zero under constant light illumination. Because these reverse photodiodes have different response speeds, the device only responds near a certain frequency, which can be tuned by manipulating the 2D perovskite components. The target device achieves an ultrafast response of 19.7/18.3 ns in the frequency-selective photoresponse range 0.8–9.7 MHz. This anti-interference photodetector can accurately transmit character and video data under strong light interference with a source intensity of up to 454 mW cm$^{-2}$.

Optical communications are attracting considerable attention for data transmission and collection, which typically require light emission, transport channels, and photodetectors[1–4]. However, not all optical communications applications have sealed light-transmission channels. As an important aspect of optical communications, free-space coupling plays an indispensable role in air-to-ground information exchange, underwater wireless optical communications, distributed light fidelity (Li-Fi) communications, and even laser-based radar signal detection[5–7]. Limited by factors such as air scattering and solar radiation interference, the signal is considerably weakened when the light signal is transmitted to the target detector while being accompanied by considerable wide-spectrum interference. Although wavelength or frequency selection technology is commonly used to solve this problem[8–17], wavelength-selective strategies cannot fully function in the surrounding broad-spectrum interference, and frequency-selective photoresponse appears to be more adaptable for extracting target signals[18,19]. As space-coupled optical communications applications, air-to-ground and distributed Li-Fi communications have

a high demand with respect to cost and system complexity. Unfortunately, the frequency selectivity always relies on a peripheral circuit, which inevitably increases the receiver's volume and complexity, hinders progress towards integration, and does not satisfy the payload and cost requirements for air-to-ground and distributed Li-Fi communications. Therefore, signal selectivity needs to urgently be achieved without system integration.

In this work, we report frequency-selective photodetectors composed of two photodiodes with a back-to-back architecture using a 2D–3D–2D-structured perovskite film. The photodiodes exhibit different response speeds, and the net current at different frequencies depends on the sum of the two reverse current values. Owing to the vertical 'V'-shaped potential distribution of the perovskite, a frequency-selective photoresponse ranging from 0.8 to 9.7 MHz (the central frequency is 3.0 MHz) is achieved. Moreover, at the bottom of the 'V'-shaped energy band, accumulated electrons considerably accelerate the recombination to achieve an ultrafast response shorter than 20 ns in a 3 × 3 mm$^2$ active area. Due to the frequency-selective

[1]School of Physical Science and Technology, Jiangsu Key Laboratory of Thin Films, Center for Energy Conversion Materials & Physics (CECMP), Soochow University, Suzhou 215006, China. [2]Institute of Functional Nano and Soft Materials Laboratory (FUNSOM), Jiangsu Key Laboratory for Carbon-Based Functional Materials & Devices, Soochow University, Suzhou 215123, China. ✉e-mail: hxsun@suda.edu.cn; lli@suda.edu.cn

photoresponse and rapid response, character and video data are transmitted in real time, even under strong interference from a light-emitting diode (LED) operating at a source intensity of 454 mW cm$^{-2}$. Our approach demonstrates the frequency-selective photoresponse in a single back-to-back-structured device without external system integration, providing promising application for space-coupled optical communications.

## Results

### Design principles of the devices

As shown in Fig. 1a, by designing two reverse photodiodes with different response speeds, the net current should only be output within the time interval Δt. When two photodiodes with different response speeds are connected in series back-to-back, the output of the entire device depends on the sum effect at the different frequencies (Supplementary Fig. 1). Notably, in space-coupled scenarios, most interfering light sources are either constant or low-frequency signals driven by solar radiation or power lines. Under constant incident light, if the

potentials generated by the two reverse photodiodes with the same magnitude but opposite directions can be accurately controlled, the external circuit will exhibit a zero-current signal. When low-frequency (less than central frequency) interference is incident, the current amplitude is considerably suppressed, although the device current cannot achieve an absolute-zero response. At the central frequency, the slow photodiode only outputs a lower-amplitude signal, while the fast photodiode can still reach the maximum output. Therefore, the maximum net output can be reached. Compared with the effective signal transmission around the central frequency, the low-frequency perturbation generated by interfering light is negligible. At higher frequencies (greater than the central frequency), neither photodiode can respond in time, which decreases the net current amplitude until it reaches zero again. The device properties, such as the central frequency, can be controlled by adjusting the distribution and intensity of the photodiodes' built-in fields. In previous work, our group found severe phase separation in 2D perovskites[20], and the phase-component distribution could be controlled using different 2D salts[21], solvents[22],

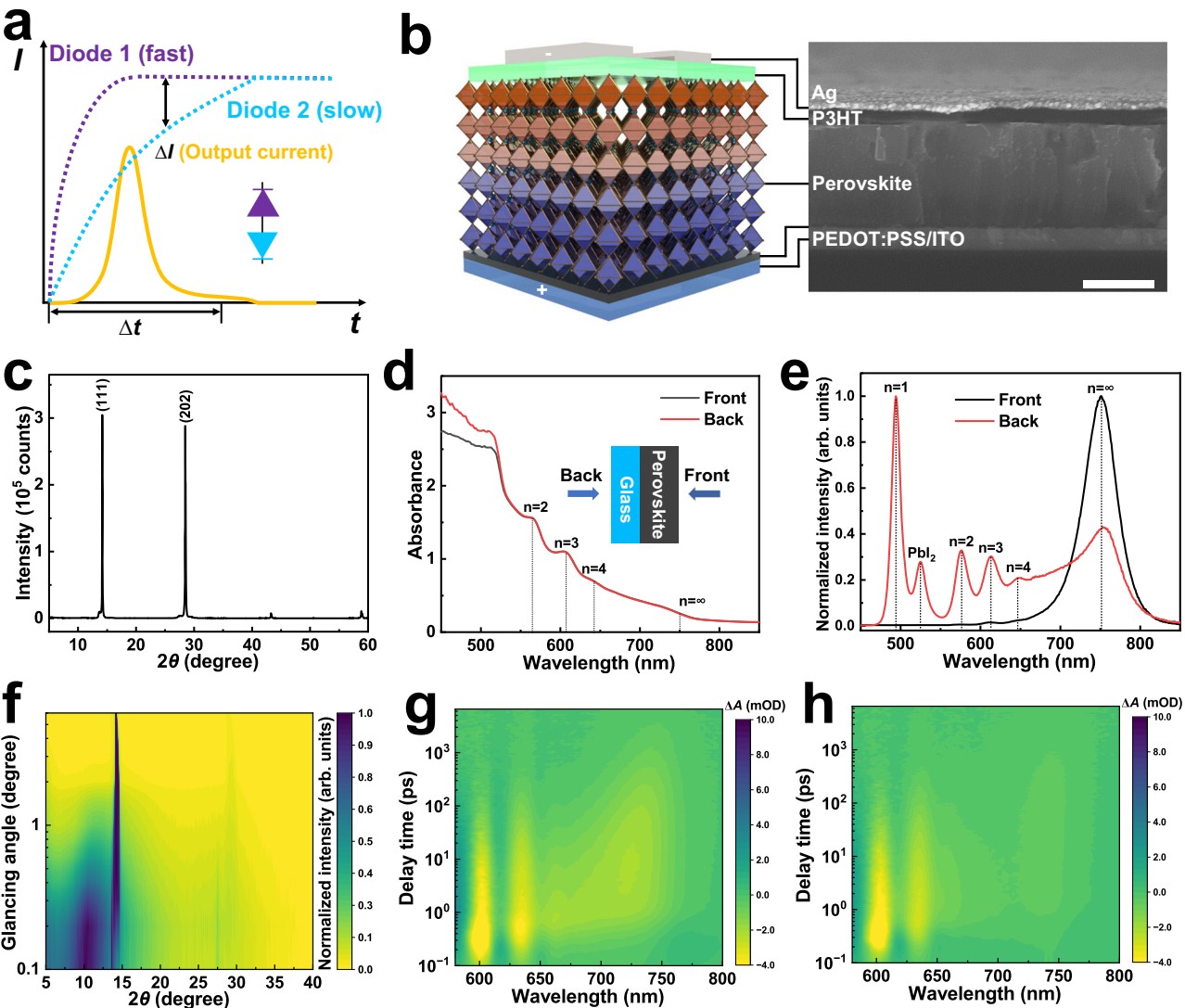

**Fig. 1 | Device structure and characterization of perovskite film. a** Graph of current *versus* time for two back-to-back-structured photodiodes exhibiting different response speeds. **b** Schematic of the device structure and corresponding cross-sectional scanning electron microscope (SEM) image. The red and blue regions indicate the front and back parts of the perovskite layer, respectively. The scale bar is 500 nm. X-ray diffraction pattern (XRD) **c**, absorption **d**, and photoluminescence (PL) spectra **e** for the (tBBA)$_2$MA$_2$Pb$_3$I$_{10}$ 2D perovskite film. **f** Grazing incident X-ray diffraction (GIXRD) 2D mapping for the 2D perovskite film from 0.1° to 6°. Time-wavelength-dependent transient absorption (TA) color maps generated from the **g** front and **h** back sides for the 2D perovskite film deposited on a glass substrate and pumped at 470 nm.

and preparation methods[23,24]. Highly oriented (tBBA)$_2$MA$_2$Pb$_3$I$_{10}$ ((tBBA)$_2$MA$_{n-1}$Pb$_n$I$_{3n+1}$, $n$ = 3 (defined as N3), where tBBA$^+$ and MA$^+$ are 4-*tert*-butylphenylmethylammonium and methylammonium ions, respectively) 2D perovskites can be prepared using simple one-step solution spin-coating assisted by the hot casting method[24]. This simple but effective method for regulating the perovskite-component distribution is highly suitable for controlling the vertical built-in fields of photodiodes. As shown in Fig. 1b, the complete device architecture comprises an approximately 700 nm–thick heterogeneous photoactive N3 perovskite layer sandwiched between two hole-transport layers (HTLs), including poly(3,4-ethylenedioxythiophene):poly(styrene sulfonate) (PEDOT:PSS) and poly(3-hexylthiophene-2,5-diyl) (P3HT). Notably, the perovskite, PEDOT:PSS, and P3HT thicknesses are critical for achieving a frequency-selective photoresponse.

## Characterization of the perovskite absorber

Figure 1c shows the X-ray diffraction (XRD) pattern of the N3 perovskite film. The strong diffraction peaks at 14.2° and 28.5° correspond to the (111) and (202) crystal planes, respectively, and their presence indicates high crystallinity. The peak intensity ratio ($I_{(202)}/I_{(111)}$) is 0.95, demonstrating that the 2D perovskite grows perpendicular to the substrate[24]. The absorption spectra in Fig. 1d show the coexistence of 2D perovskites ($n$ = 2, 3, and 4) and a 3D perovskite ($n$ = ∞) from the perovskite/air (front) side to the perovskite/glass (back) side, and the perovskite-phase distribution needs to be further analysed. As shown in Fig. 1e, only 3D perovskite emission peaks appear in the photoluminescence (PL) for the front side, while numerous 2D perovskite emission peaks appear for the back side; these results indicate that the 2D perovskite is accumulated at the back and that the 3D perovskite is accumulated at the front. However, the front part of the film is not solely composed of the 3D perovskite[20,22,23]. The absence of the PL signals for the film's 2D perovskite components is likely due to efficient carrier separation on the front side. The film's grazing-incidence XRD (GIXRD) pattern exhibits additional characteristic peaks at around 10.2°, 13.6°, and 27.5° at low glancing angles, as shown in Fig. 1f and Supplementary Fig. 2. According to the calibration of the (tBBA)$_2$PbI$_4$ (N1), (tBBA)$_2$MAPb$_2$I$_7$ (N2) (Supplementary Fig. 3), and N3 (Fig. 1c) XRD patterns, the peaks at 10.2° and 13.6° correspond to the N2-phase component, and the peak at 27.5° corresponds to the N3-phase component; these results indicate that the 2D perovskite is enriched on the film's front surface. To further analyse the component distribution along the vertical direction, grazing incidence wide angle X-ray scattering (GIWAXS) was performed at grazing angles of 0.1° and 0.5°, as depicted in Supplementary Fig. 4. The discrete Bragg spots indicate that the 2D perovskite has good crystalline orientation. At a low incidence angle of 0.1°, the Debye-Scherrer diffraction ring at $q_z$ = 0.22 Å$^{-1}$ shows the (0$k$0) crystal plane of the 2D perovskite (Supplementary Figs. 4a and 5a). The diffraction peaks derived from GIWAXS exhibit a similar trend to those from GIXRD, particularly at approximately 14° and 28° (Supplementary Fig. 5b and 5c). Moreover, with increasing incidence angle, the intensity of the 2D perovskite peak remains practically unchanged, while the intensity of the 3D perovskite peak significantly increases. This observation implies that the middle region is predominantly composed of 3D perovskite, while the surface region is enriched with 2D perovskite. From the GIWAXS, GIXRD, and PL spectra, the perovskite film exhibits a vertical 2D–3D–2D component distribution. To further confirm the phase-component distribution and investigate carrier transport, femtosecond transient absorption (TA) measurements are conducted. As shown in Fig. 1g and 1h, a 470 nm–wavelength pump light with a photon energy higher than that of the bandgap of all phase components is utilized to irradiate the front and back sides of the perovskite film deposited on the glass substrate. The distinctive negative ground-state bleach (GB) bands at 599, 634, and 722 nm are assigned to populations of excited charge carriers in the $n$ = 3, 4, and ∞ phase components on both sides of the film,

respectively; thus, 2D perovskite is present on both sides of the film[23]. The peak corresponding to the 3D perovskite barely appears in the TA spectrum for the film's back side, which further confirms the extensive accumulation of 2D-phase components in the film's backside. The TA spectra measured at different delay times are shown in Supplementary Fig. 6. The GB peaks corresponding to the $n$ = 3 and 4 phase components rapidly reach the maximum intensity within 1 ps and then gradually weaken, while the 3D GB peak is intensified; these results indicate that carriers generated in 2D components are transferred to the 3D perovskite[25]. The carrier transport from the 2D components to the 3D perovskite is clearly observed in the spectra of both sides of the film. Therefore, through the combination of PL, GIXRD, and TA characterization, the 2D–3D–2D phase distribution is confirmed and used to construct two reverse photodiodes, which provides the foundation for generating a frequency-selective photoresponse.

## Optimization of the frequency-selective photoresponse

The key for achieving a frequency-selective photoresponse is to regulate the electric field by controlling both the thickness of each layer and the phase composition. The layer thickness can be finely adjusted by changing the precursor concentration. As shown in Supplementary Fig. 7, the external quantum efficiency (EQE) over the entire wavelength range is below 30% owing to the competition between both reverse photodiodes; this indicates a low photoresponse under the illumination of constant monochromatic light. This competition provides the device with bipolar response characteristics and reduces the amount of steady-state current generated by the solar spectrum. Under AM 1.5 G irradiation, the integrated currents are all below 3 mA cm$^{-2}$ for the devices fabricated with different thicknesses (Supplementary Fig. 8). The use of the PEDOT:PSS stock solution diluted 1:1 ($v$:$v$) with water, 0.8 M perovskite precursor solution and 20 mg mL$^{-1}$ P3HT achieve a minimum EQE of < 1.5% in the visible range, and the corresponding integrated current under AM 1.5 G irradiation is as low as 0.0925 mA cm$^{-2}$ (Fig. 2a). A 450 nm–wavelength laser is utilized as the communication source. At a low frequency of 1 Hz, the device does not generate current (Fig. 2b). Because the central frequency is closely related to the built-in electric field strength (i.e., relative positions of the perovskite and HTL Fermi levels) and carrier mobility[26–28], the alteration of the functional layer thickness within a certain range does not directly impact the position of the central frequency (Supplementary Fig. 9). However, the unoptimized HTLs and perovskite layer thickness can cause a significant leakage current in the low-frequency region. For instance, irrespective of how the thickness of the P3HT layer is altered, the central frequency always locates at 500 kHz, as shown in Supplementary Fig. 9b. However, if the thickness of the P3HT layer is not finely adjusted to balance the two built-in electric fields, a significant response is observed in the low-frequency region less than 100 Hz, and this response is completely detrimental to anti-interference implementation. Therefore, to adjust the central frequency, the 2D ammonium salt material needs to be modified. The type of 2D ammonium salt considerably affects the central frequency. Ethylammonium iodide (EAI), butylammonium iodide (BAI), phenylmethylammonium iodide (PMAI), phenylethylammonium iodide (PEAI), octylammonium iodide (OAI), decaneammonium iodide (DAI), and 4-tert-butyl-benzylammonium iodide (tBBAI) with different chain lengths and functional groups are selected as appropriate 2D ammonium salts for modulating the central frequency. The BAI-, PEAI-, and tBBAI-based perovskite films show good crystallinities and crystal orientations (Supplementary Fig. 10a), corresponding to the uniform and dense perovskite film morphology shown in Supplementary Fig. 11. Except for the EAI-modified perovskite, all other 2D-ammonium-salt-modified materials present absorption peaks for $n$ = 2, 3, and 4 phase components, indicating pervasive phase separation (Supplementary Fig. 10b). Furthermore, the PL spectra measured on the films' front and back sides show that the phase components

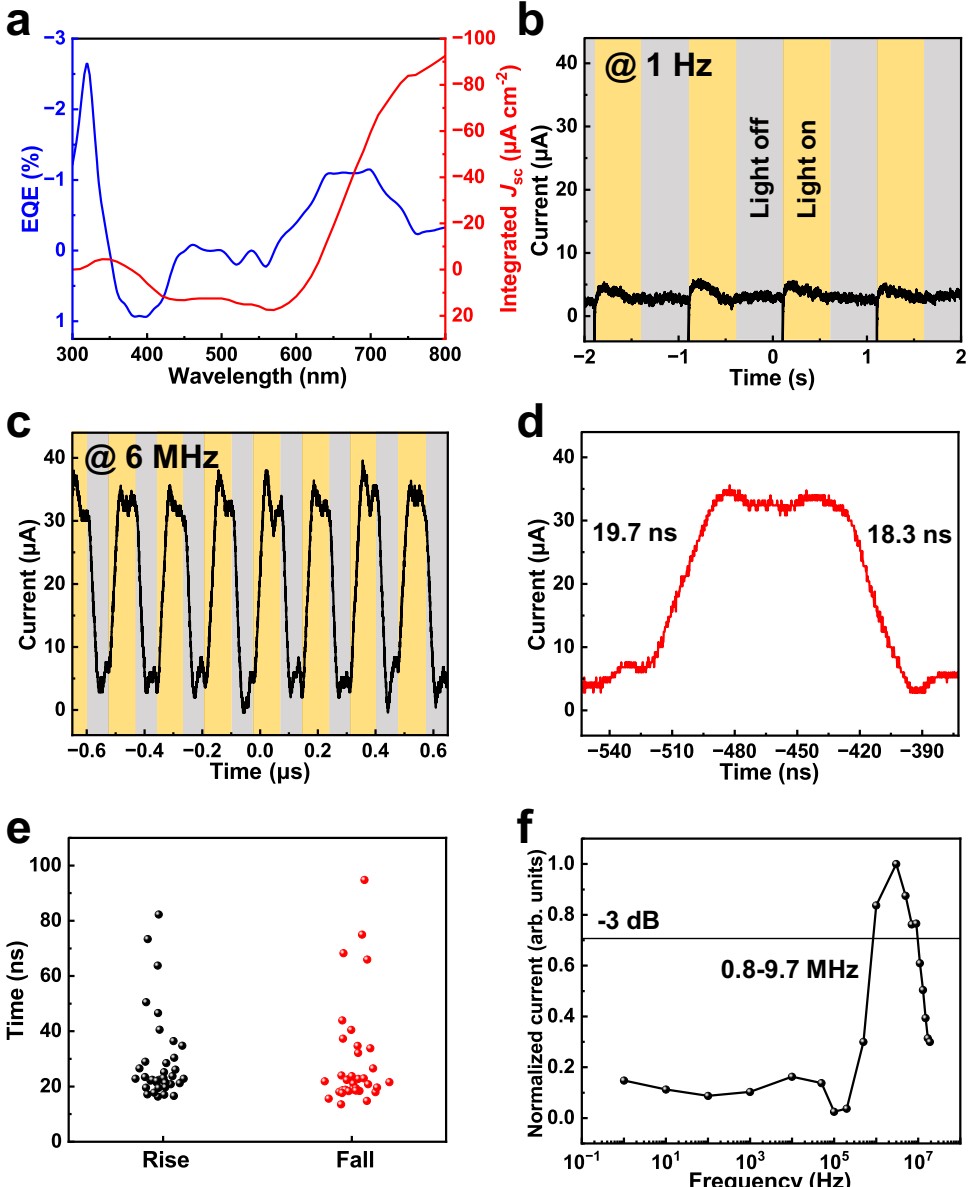

**Fig. 2 | Anti-interference and frequency-selective photoresponse detection.** **a** External quantum efficiency (EQE) and integrated current density ($J_{sc}$) of the functional device. Current *versus* time measured at 1 Hz **b** and 6 MHz **c**. **d** Enlarged pattern from **c**. **e** Statistical graph of the rise and fall time for 36 devices. **f** Normalized response plotted as a function of the input-signal frequency, indicating a −3 dB cut-off frequency.

along the vertical direction are unevenly distributed (Supplementary Fig. 12). To investigate the universality of this strategy, the same 2D-3D-2D phase distribution was examined in PEAI and BAI, as evidenced by GIXRD (Supplementary Figs. 13 and 14) and the PL spectra (Supplementary Fig. 12). As shown in Supplementary Fig. 15, the 2D-ammonium-salt-based devices exhibit distinct −3 dB cut-off frequencies. Devices based on MAI, PMAI, and BAI exhibit a loss of frequency-selective response characteristics due to significant leakage current in the low-frequency region. In contrast, devices based on EAI, PEAI, and tBBAI demonstrate more pronounced frequency-selective characteristics. The central frequency of EAI-based devices is only 0.2 MHz. Moreover, due to slow rise/fall times of 0.3/0.5 µs, which are much slower than those of the tBBAI-based device, the response bandwidth of the PEAI-based device is below 1 MHz (Supplementary Fig. 16). Therefore, the 2D ammonium salt can be modified to adjust the central frequency over a wide range. Owing to the fastest response speed, tBBAI is selected as the 2D ammonium salt for fabricating the final frequency-selective device for application to optical

communications. The phase components are more finely regulated by adding different concentrations of tBBAI to the precursor solution. The N3 and (tBBA)$_2$MA$_3$Pb$_4$I$_{13}$ (N4) perovskite films exhibit uniform pinhole-free morphologies (Supplementary Fig. 17). The devices fabricated using N3 and N4 perovskite films achieve rapid frequency-selective photoresponse (Supplementary Fig. 18) with slightly different response speeds. Thus, the N-value regulation can be used for fine-tuning the central frequency. Based on the previous XRD measurements (Fig. 1c and Supplementary Fig. 3) and because N3 has a better crystal orientation than N4, N3 is selected as the photoactive layer. At a high frequency of 6 MHz, the N3 perovskite device generates a photocurrent with an amplitude of 25.6 µA (Fig. 2c), while at a low frequency, the photocurrent approaches 0 (Fig. 2b). The enlarged pattern in Fig. 2d shows fast edges of 19.7/18.3 ns, indicating rapid signal transmission. Figure 2e shows the statistical distributions for the rise and fall time for 36 devices. All devices rapidly respond within 100 ns, and most response times are approximately 20 ns. As shown in Fig. 2f, the −3 dB cut-off frequency is calculated in the range 0.8−9.7 MHz,

which satisfies the frequency range requirement for free-space optical communications.

## Charge carrier dynamics and energy band

To investigate the effects of the back-to-back device structure on the vertical electrical field and carrier distribution behavior, cross-sectional Kelvin probe force microscopy (KPFM) is used to profile and observe the variations in the contact potential differences (CPDs) measured under dark and light conditions. By calculating the first and second derivatives of the CPD, the vertical electric field and charge-density distribution profiles are determined, as shown in Fig. 3a–h[29,30]. When the device is illuminated, the CPD decreases for the entire perovskite layer, indicating that photogenerated electrons are trapped in the perovskite layer. This result can be attributed to the intrinsic 2D-3D-2D phase structure of the perovskite, which generates a 'V'-shaped built-in electric field. The electron-blocking materials (P3HT and PEDOT:PSS) on either side of the perovskite layer further decrease the CPD, leading to the enhanced trapping of the photoelectrons within the perovskite film. The resulting electron accumulation within the perovskite layer provides the basis for the rapid decrease in the light response edge. Notably, the CPD shift is particularly pronounced at the P3HT–perovskite interface; this result indicates that this region functions as the primary electron-storage site following the competition between both reverse potentials. Notably, under both dark and light conditions, the PEDOT:PSS side exhibits a larger potential difference than the P3HT. When the device is illuminated, the vacuum-level change ($\Delta E_{vac}$) indicates that the junction field on the P3HT side nearly vanishes; thus, the PEDOT:PSS/perovskite junction field becomes dominant (Supplementary Fig. 19). When the device is in the dark, two prominent peaks appear in the spectra for both HTL–perovskite interfaces, and no flat electric field plateau is observed; therefore, the perovskite is completely depleted[29]. As shown in Fig. 3g, the holes clearly accumulate on the front surface of the perovskite and at the perovskite/PEDOT:PSS interface, and the electrons mainly accumulate in the middle of the perovskite. The charge carrier accumulation results from the potential well at the heterointerface and inside the

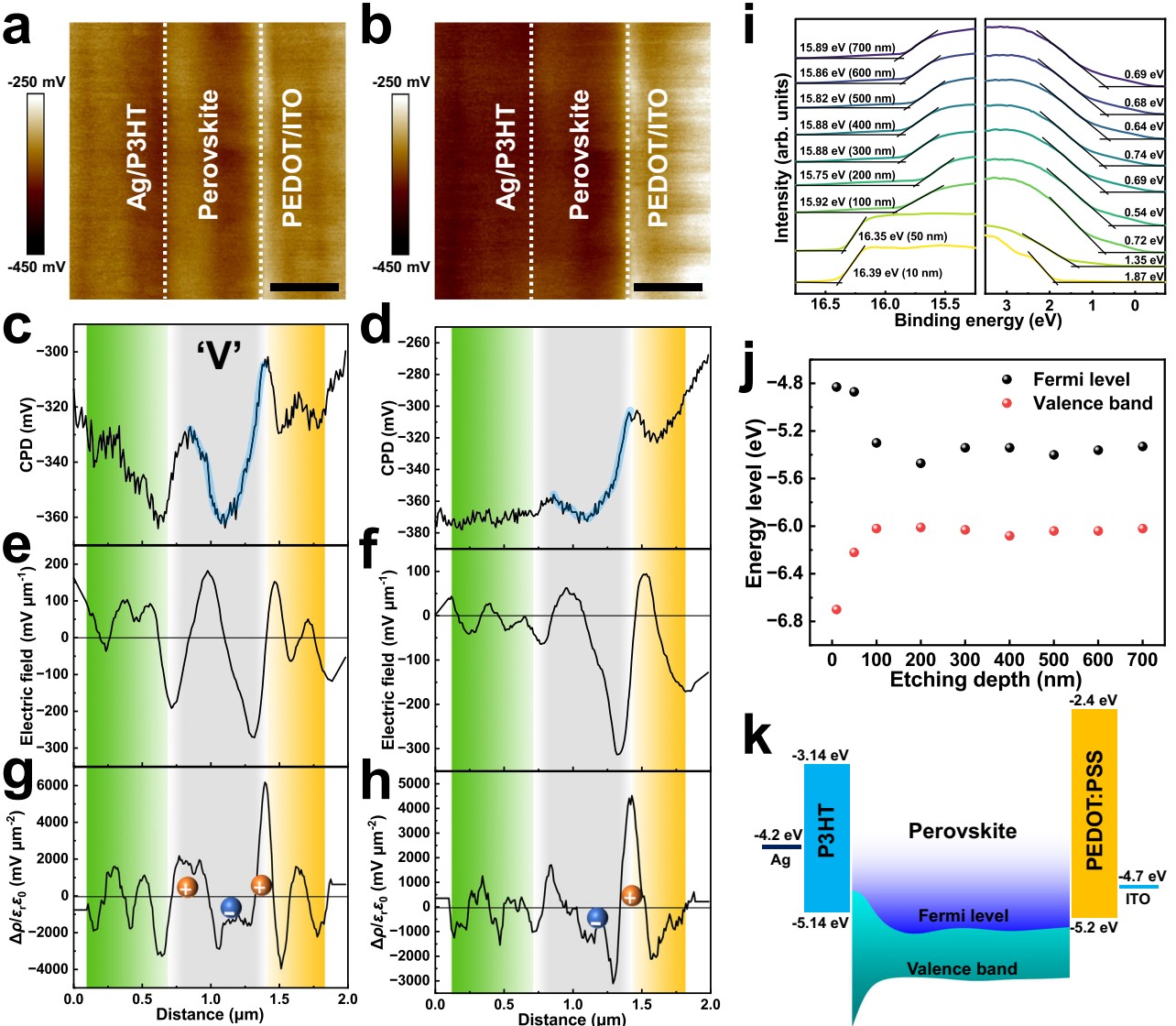

**Fig. 3 | Perovskite and heterointerface dynamics.** Cross-sectional Kelvin probe force microscopy (KPFM) images of the Ag/P3HT/perovskite/PEDOT:PSS/ITO device under dark **a** and light **b** conditions. The scale bar is 500 nm. Contact potential differences (CPDs) measured under dark **c** and light **d** conditions. Electric field difference obtained by calculating the first derivatives of **c** and **d** under dark **e** and light **f** conditions. Charge-density distribution profiles obtained by calculating the second derivatives of **c** and **d** under dark **g** and light **h** conditions. **i** Ultraviolet photoelectron spectroscopy (UPS) spectra for N3 perovskite films etched at different depths. **j** Energy-level diagram for N3 perovskite films etched at different depths. **k** Energy-band diagram for the back-to-back-structured device.

perovskite layer that traps charges, as evidenced by the $\Delta E_{vac}$ values in Supplementary Fig. 19 and ultraviolet photoelectron spectroscopy (UPS) results shown in Fig. 3i and j[30]. Based on the band alignments deduced from the UPS spectra, the photogenerated electrons are confined in the perovskite layer due to the electron-blocking layers on either side. More specifically, these electrons are located closer to the P3HT side because of their intrinsically higher Fermi level; this result agrees with the KPFM analysis. Notably, the accumulation of the 2D ammonium salt on the surface causes the anomalous Fermi energy level at the front side[20]. This will lead to spontaneous electron extraction behavior. For this band structure, during dark–light transitions, photons enter from the indium tin oxide (ITO)/PEDOT:PSS side and excite the highly oriented 2D/3D hybrid perovskite. The photogenerated holes are rapidly extracted by PEDOT:PSS and travel through the ITO electrode with the shortest transport distance, resulting in a fast rise edge. Although P3HT can extract holes, this effect is considerably reduced by the low valence band position on the perovskite surface and the reverse field generated by the higher Fermi level on this side. P3HT primarily serves as an electron-blocking layer. In short-term illumination, electrons slightly accumulate within the perovskite, the reverse field disappears (as indicated by the flat P3HT interfacial potential when the device is illuminated), and P3HT begins counteracting the electromotive force on the PEDOT:PSS side. When the device is constantly illuminated, the current ultimately returns to zero, satisfying the original design objective. During rapid operation and light–dark transitions, the device current is dominated by the built-in electric field on the PEDOT:PSS side because P3HT does not effectively extract holes.

When the N3 perovskite is directly deposited on an ITO substrate, the lack of electron-blocking layers on both sides considerably reduces the carrier accumulation, as shown in Supplementary Fig. 20. However, owing to the 3D–2D structure on the ITO side, the holes can still be extracted with the aid of the 2D perovskite, leading to observable hole accumulation at the ITO interface when the device is illuminated. This 3D–2D structure is often beneficial for achieving efficient hole transport[31–33]. When PEAI perovskite is utilized in the photoactive layer, the 'V'-shaped band structure still appears and ensures a basic frequency-selective photoresponse. However, due to the suboptimal band structure, the accumulation of the photogenerated electrons is not pronounced (Supplementary Fig. 21). Due to the absence of a potential well, the 3D $MAPbI_3$ perovskite cannot even accumulate enough electrons to generate a frequency-selective photoresponse (Supplementary Fig. 22).

## Characteristics of the half-device

To clarify and verify the respective roles of the HTL materials on both sides of the device in carrier transport, half-devices fabricated with P3HT/perovskite/ITO and perovskite/PEDOT:PSS/ITO structures are characterized. The PL peak intensity is used to determine the radiative recombination degree of carriers and indirectly indicates the number of carriers. By measuring the PL spectra for the front and back sides of the single-HTL and HTL-free devices, the carrier accumulation or transmission can be characterized in a single junction. As shown in Supplementary Fig. 23a, only the 3D perovskite emission peak appears in the PL spectra when the devices are irradiated from the front side. The difference is as follows: P3HT intensifies the emission peak, while PEDOT:PSS weakens the emission peak. The emission peak intensification occurs because P3HT offsets the strong electron extraction provided by phase separation at the front side, enabling more carriers to remain there for radiative recombination. The hole extraction by PEDOT:PSS at the back side reduces the number of carriers available for radiative recombination, which weakens the emission peak. When the device is irradiated from the back, P3HT still prevents spontaneous electron extraction at the front side; this considerably intensifies both the 2D and 3D emission peaks (Supplementary Fig. 23b). However,

PEDOT:PSS can extract holes from adjacent 2D perovskite layers, weakening the emission peak for the corresponding 2D components. The 3D perovskite emission peak intensification is attributed to the transfer of the carriers generated by illumination excitation from the 2D perovskites to the 3D component. Additionally, this phenomenon is confirmed by the TA spectra. As shown in Supplementary Figs. 24 and 25, PEDOT:PSS does not change the TA spectra compared with that for the pure perovskite deposited on the glass substrate, as shown in Fig. 1g and h, respectively; these results can be attributed to the consistency between the hole extraction behaviors of the PEDOT:PSS and 2D perovskite components and, thus, does not prevent the original carrier-transport behavior. However, P3HT considerably alters the carrier dynamics, particularly by extending the lifetime of the $n = 4$ and 3D perovskites because P3HT inhibits the electron self-conduction to the perovskite front surface; this result is consistent with the PL spectra. The cross-sectional KPFM images show the detailed carrier accumulation (Supplementary Figs. 26 and 27). Clearly, P3HT is crucial for achieving the 'V'-shaped energy band. Due to the inherent electron extractability of the perovskite front surface, the P3HT-free device barely accumulates photogenerated carriers. In contrast, for the PEDOT:PSS-free device, the 'V'-shaped energy-band structure is still partially maintained because the wide-bandgap 2D components are widespread at the back surface. However, because the distribution of the built-in electric field is disrupted, the leakage current should considerably increase under constant illumination. The half-device photoresponse is shown in Supplementary Fig. 28. Clearly, at low frequencies, the absence of a unilateral HTL causes the device to exhibit a high photogenerated current, which is not conducive for achieving a frequency-selective photoresponse. Therefore, the perovskite thickness needs to be optimized, and dual HTLs need to be applied to the device to establish a competitive built-in electric field between the front and back. When these two fields are balanced, the current approaches zero at low frequencies. Notably, when the P3HT layer is absent, the device's response speed considerably decreases. The absence of the P3HT layer, wherein electrons primarily accumulate on one side of the device, causes the device to lose its 'V'-shaped energy band. Therefore, carrier transport and recombination both revert to their traditional mechanisms in perovskite photodetectors. On the PEDOT:PSS side, wide-bandgap 2D perovskite components are widespread in the perovskite and slightly maintain the 'V'-shaped electric field. Hence, the PEDOT:PSS-free device retains a rapid response. Additionally, because the competition between the built-in electric fields on both sides is designed to equilibrate throughout the entire device, the current can be reversed when the HTL is removed from one side of the device.

## Demonstration of anti-interference communications

In practical applications, free-space optical communications systems consist of signal sender and receiver units (Fig. 4a). The transmitted signal is encoded and converted to a series of zeros and ones in the American Standard Code for Information Interchange (ASCII), and the laser diode's switching state is controlled for transmitting the signal. Then, the photodetector receives the signal and generates the corresponding light and dark currents; these are restored to the data signal by the driver and finally restored by the computer. The ASCII character codes (ECS, Soochow, and University) are shown in Fig. 4b. The widely reported perovskite photodetector ITO/$SnO_2$/$MAPbI_3$/Spiro-OMeTAD/Ag (n-i(3D)-p type) was selected as the control device[34,35]. The XRD and PL characterizations in Supplementary Fig. 29 demonstrate its good crystal quality. In the presence of external interference, the signals show distortion (Supplementary Movie 1). Our device can still accurately receive signals under interference from an LED light operating at a source intensity of 170 mW cm$^{-2}$; however, the signal appears garbled from the interference caused by the LED operating at a source intensity of 454 mW cm$^{-2}$ (Supplementary Movie 2). For comparison,

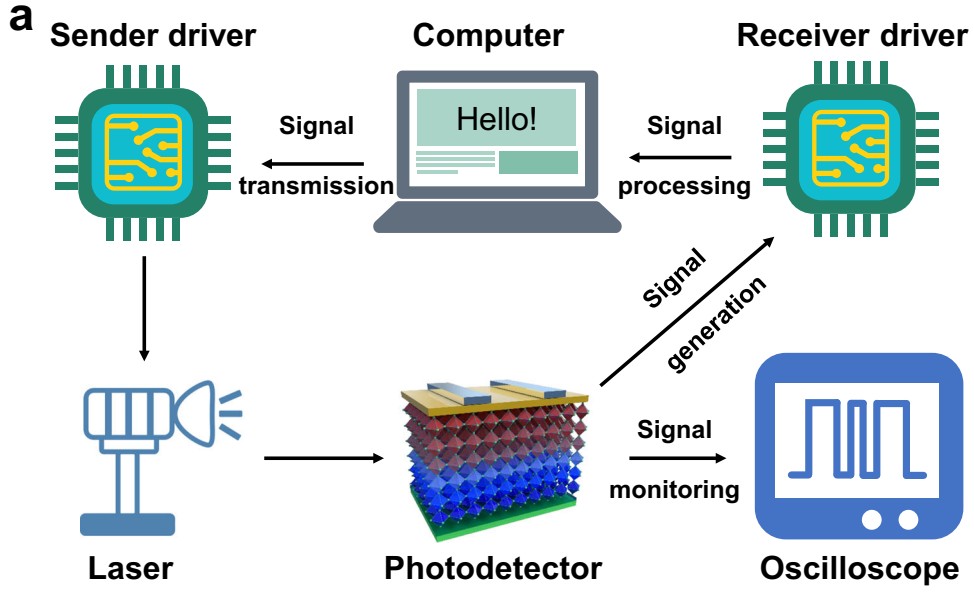

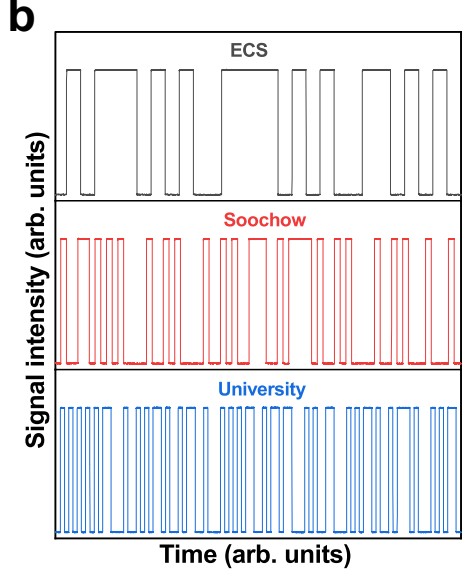

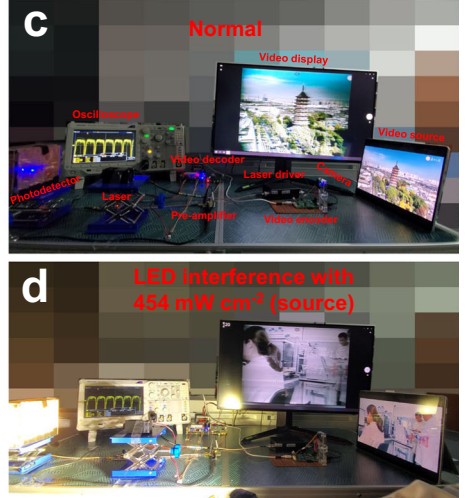

**Fig. 4 | Demonstration of free-space optical communications. a** Schematic of the free-space optical communications system. **b** Laser driving signals during character transmission. Photos of the video data transmitted in **c** an indoor light environment and **d** intense light interference generated by light emitting diode (LED). The encoding method for video transmission is a phase alternating line (PAL), and the laser intensity irradiated on the photodetector is 5.84 mW cm$^{-2}$.

the LED light-source spectrum is shown in Supplementary Fig. 30. To confirm the necessity of the 'V'-shaped built-in electric field, MAPbI$_3$ was configured as an ITO/PEDOT:PSS/MAPbI$_3$/P3HT/Ag (p-i(3D)-p type) structure device (Supplementary Movie 3). This p-i(3D)-p-type device could barely withstand the interference of 55.1 mW cm$^{-2}$ background light. In the presence of the blocking layers on both sides, the perovskite could be approximated as the base of the 'V'-shaped band; however, at this junction structure, the relative relationship between the band of the CTL and the perovskite was established, lacking a variable to balance the two intrinsic electric fields. Therefore, the P3HT electron blocking layer on the upper surface is crucial; when N3 perovskite was configured as an ITO/PEDOT:PSS/N3 perovskite/PCBM/Ag structure device, its anti-interference ability completely disappeared (Supplementary Movie 4). In the absence of a blocking layer, the perovskite's 'V'-shaped band structure was incapable of charge accumulation. These videos confirm the importance of the dual HTL structure and the selection of 2D-3D-2D perovskite. Because the information

density for video data is higher than that for character data, video data transmission requires a higher frequency than character data transmission. Our devices have sufficiently fast response and completely satisfy the requirements of high-density information transmission (Fig. 4c, d). In the interference generated under constant illumination with a low-frequency LED operating at source intensities of 55.1, 170, and 454 mW cm$^{-2}$, the device can still accurately transmit video signals. When the light-source intensity finally reaches 910 mW cm$^{-2}$, the video signal begins to distort; this result indicates that the anti-interference signal transmission capability of the device is strong (Supplementary Movie 5). Significantly, in the presence of flickering interference, subtle video jitter was observed. As depicted in Supplementary Fig. 31, the square wave interference undergoes wavelet transformation, showing a multitude of high-frequency components in the MHz range. These high-frequency signals are subsequently captured by detectors, leading to interference in the video transmission process. As evidence, when pure low-frequency sinusoidal light is used as the interference

signal, no disruption to the signal occurs (Supplementary Movie 6). Owing to the insufficient response speed of the devices with conventional structures, a commercial high-speed silicon photodetector (model: S6968, Hamamatsu) was utilized as a reference. Although capable of video transmission, even slight external disturbances result in instantaneous signal loss, as shown in Supplementary Movie 7.

## Discussion

In this work, we have fabricated a back-to-back (Ag/P3HT/perovskite/PEDOT:PSS/ITO) structure to counteract the current of two photo-diodes, and the resulting single device achieves a frequency-selective photoresponse. Through simple one-step spin-coating, the 2D perovskite $(tBBA)_2MA_2Pb_3I_{10}$ shows a vertical 2D–3D–2D phase-component distribution. In addition to the thickness regulation of the double HTLs, the front- and back-field intensities are balanced to eliminate the photoresponse when the device is constantly irradiated. The device exhibits a fast response of 19.7/18.3 ns in the response range 0.8–9.7 MHz. Despite the interference generated by an LED light, video data can still be accurately transmitted, demonstrating the device's good anti-interference transmission capability. By replacing integrated systems with a single device, the complexity and costs can be considerably reduced, which enables greater flexibility and diversity for designing free-space optical communications equipment.

## Methods

### Materials

tBBAI, MAI, P3HT, and PEDOT:PSS4083 solution were purchased from Xi'an Yuri Solar Co., Ltd. Lead iodide ($PbI_2$) was purchased from TCI Chemicals. *N,N*-dimethylformamide (DMF) and chlorobenzene (CB) were purchased from Sigma–Aldrich. All chemicals were used as received without further purification.

### Preparation of perovskite precursor

The perovskite precursor concentration was defined by the $PbI_2$ content. For example, 0.9 M $(tBBA)_2MA_2Pb_3I_{10}$ was prepared by dissolving 0.6 mmol of tBBAI, 0.6 mmol of MAI, and 0.9 mmol of $PbI_2$ in 1 mL of DMF. The concentrations of the other ammonium salts were controlled at 0.8 M. The precursor solutions were stirred in a glovebox overnight and filtered before use.

### Device fabrication

The photodetectors were fabricated on commercial ITO-patterned glass electrodes. A 3 mm wide channel was etched into the ITO electrodes, and they then were cleaned by sequential sonication in acetone, alcohol, and deionized water for 30 min each. The cleaned substrates were treated with both ultraviolet (UV) light and ozone for 15 min before being spin-coated with different concentrations of the PEDOT:PSS solution at 5000 rpm for 30 s. The PEDOT:PSS-coated substrates were heated on a hot plate at 150 °C for 30 min and then cooled to room temperature. Then, the substrates were transferred to a $N_2$-filled glovebox without any further UV treatment. The substrates were preheated at 110 °C for 5 min and then quickly transferred to a spin-coater as soon as possible. Then, within 3 s, 50 μL of the perovskite precursor was spin-coated on the preheated substrates at 5,000 rpm for 20 s without ramping. All other 2D ammonium salt-based devices were fabricated in the same manner, except OAI and DAI, whose strong hydrophobicity hindered the adhesion of HTL. The coated substrates were then heated at 100 °C for 10 min, and 40 μL of the P3HT solution (different concentrations dissolved in CB at 70 °C) was deposited on the perovskite surface at 3000 rpm for 30 s. Finally, 90 nm–Ag electrodes were thermally evaporated using a shadow mask.

### Characterizations

The surface morphologies of the perovskite films and device cross-sectional structures were recorded using field-emission scanning electron microscopy (FESEM, SU8100, Hitachi). XRD and GIXRD patterns were measured using an X-ray diffractometer equipped with a Cu Kα radiation source (D8-Advance, $\lambda = 0.154$ nm) and a LYNXEYE XE-T detector. The GIWAXS test was performed at beamlines BL03HB at the Shanghai Synchrotron Radiation Facility (SSRF). Optical absorption spectra were collected using an ultraviolet–visible (UV–vis) spectrophotometer (Shimadzu, UV-3600). The PL spectra were measured using an Edinburgh FLS 980 instrument at room temperature. The excitation source was 450 nm monochromatic light emitted from a xenon lamp. For femtosecond TA spectroscopy, a 200 fs, 470 nm–wavelength pump pulse was used to excite all the samples, and the probe beam was detected using a spectrometer from 440 to 860 nm. The EQE was measured using an electrochemical workstation (Autolab, PGSTAT 302 N). The detailed constant current was measured using a 0.0625 $cm^{-2}$ mask, and the monochromatic light intensity was calibrated using standard silicon. At different frequencies, the current was recorded using an oscilloscope (Tektronix, MDO3102), and the device was excited using a 450 nm–wavelength laser diode (OSRAM) driven by a signal generator. The current signal recorded by the oscilloscope was converted with a 50-ohm resistor. To enhance the accuracy of current signal measurements using an oscilloscope, all test fixtures were shielded, all connections were minimized in terms of length and quantity, and only standard SMA or BNC interfaces were used. The oscilloscope's acquisition was set to high resolution mode to improve vertical resolution. The sync signal of the light source was used as the trigger source, and the final response amplitude was defined as the difference of the average current value under the light and dark state within one cycle. A preamplifier based on OPA657 was used only during signal transmission demonstration (recording supporting videos). The video transmission method modulated the camera data into a phase alternating line (PAL) format, which was used as the driving current of the laser diode for intensity modulation after amplification. After the detector collected the current signal, it was input to the acquisition card and converted into a video streaming output to the screen. All tests were performed at 0 V bias. Cross-sectional KPFM was performed using a MultiMode 8 instrument (BRUKER) operating in noncontact mode to record topographic and CPD images with a PtIr-coated tip (SCM-PIT-V2, BRUKER). The drive routing was set at the sample, and no external bias was applied to the device. The work function of the AFM tip and the cross-section of the perovskite were calibrated using gold (Au), with a known work function of 5.1 eV. This calibration process ensured that the sample remained undamaged during preparation. In the light-state KPFM test, the light source was a xenon lamp with an intensity of approximately 554.2 mW $cm^{-2}$. The UPS measurements were conducted using a Thermo Fisher ESCALAB Xi+ instrument and 21.22 eV He I photoelectrons. Argon plasma was used to etch the perovskite films to different depths.

## Data availability

The authors declare that all data supporting the findings of this study are available within the paper and the Supplementary Information, or available from the authors upon request to H.S. (hxsun@suda.edu.cn) or L.L. (lli@suda.edu.cn).

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

## Acknowledgements

This work was supported by the National Natural Science Foundation of China (52025028, 52202273, and U22A20137), the Natural Science Foundation of Jiangsu Province (BK20210728), and the "Shuangchuang" Program of Jiangsu Province and was funded by the Priority Academic Program Development (PAPD) of Jiangsu Higher Education Institutions. We would like to thank Chen Li and Zhenzhu Wang for their support in the KPFM test. We thank the staff at BL03HB beamline of the Shanghai Synchrotron Radiation Facility (SSRF), Shanghai Advanced Research Institute, CAS, for providing technical support in X-ray diffraction data collection and analysis.

## Author contributions

L.M., H.S., and L.L. conceived the idea and designed the experiments. L.M. carried out all device fabrication and characterization. J.Z. carried out the GIWAXS test and data analysis. L.M., H.S., and L.L. composed the manuscript. L.M., H.S., L.G., M.W., F.C., and L.L. contributed to the overall scientific interpretation and revised this manuscript.

## Competing interests

The authors declare no conflicts of interest.
