## [Peer Review File · Nature Communications]

Frequency-selective perovskite photodetector for anti-interference optical communicationsREVIEWER COMMENTS

Reviewer #2 (Remarks to the Author):

In this manuscript, the authors report an asymmetric 2D–3D–2D perovskite structure to achieve a frequency-selective photoresponse, which can prohibit light interference up to 454 mW cm⁻². The concept that combining two opposite diodes in a single device with different cut-off frequencies to realize frequency-selectivity is very interesting. The manuscript has exhibited comprehensive investigations in designing the device structure and the composition of each functional layers. The demonstration of free-space communication with strong anti-interference performance is impressive. However, several concerns should be addressed in the current version as below.

(1) In the manuscript, the authors use different description of the thin film structure, including “front side” and “back side”, “lower part” and “upper part”, “upper surface”, “frontal”, “glass side”, “perovskite side”, etc., which are suggested to use uniform description and add clear labeling in Figure illustration.

(2) Please explain why there are negative EQE shown in Figure S5.

(3) The bias applied on the device when performing photodetection test should be mentioned. As described in the Experimental Section, the current was recorded using an oscilloscope. However, the oscilloscope can only record voltage signal. Did any amplifier used in the experiment?

(4) Line 150, the authors claim that “... the alteration of the functional layer thickness within a certain range does not directly impact the central frequency.” Line 92, another description “the perovskite, PEDOT:PSS, and P3HT thicknesses are critical for achieving frequency-selective photoresponse.” Do the two descriptions contradict each other? The elucidation is not clear.

(5) Are Figure 3a and b the AFM topographic images or the KPFM surface potential (SP) images? To demonstrate the surface potential under dark and illumination conditions, KPFM images are more suitable. Dashed lines should be added into KPFM images to show the position corresponding to the SP line profiles in Figure 3c and d.

(6) Was the KPFM DC bias applied on the tip or on the sample when performing KPFM? Usually, the DC bias is applied to AFM tip, and consequently the higher Fermi level (lower work function) exhibits higher surface potential. Line 199, the accumulation of electrons in perovskite will lift the Fermi level, as well as the surface potential, which is inconsistent with Figure 3c and d. Please add more experimental details and make explanations.

(7) The authors mentioned that the perovskite contains 2D-3D-2D phase separation. But why do the UPS results (Figure 3i-k) not show the change of Fermi level and valence band of perovskite at PEDOT:PSS interface similar to P3HT-Perovskite interface?

(8) The device used in free-space optical communications is ITO/SnO₂/3D perovskite/Spiro-OMeTAD/Ag, which is completely different from the one discussed above, i.e., Ag/P3HT/perovskite/PEDOT:PSS/ITO device. It is very strange to use a device never discussed above for demonstration.

(9) In the video file (445007_0_video_61698_rwpbf1) that shows the data transmission of “Soochow University”, when the light interference illuminates the receiver, the baseline of signals in oscilloscope apparently lifts, indicating that the device responds to the light interference. This is different from the mechanism that the responses of opposite diodes can cancel each other at low frequency. The frequency-dependent responsivity of photodetector should be included and discussed.

(10) Because the letters data are encoded by ASCII as digital signals, is there a comparator used for determining high (1) and low (0) levels? Is it suitable to use digital signal to prove

the anti-interference performance? Besides, more experimental details should be added, including the laser intensity irradiated on photodetector and the encoding methods for video transmission.

(11) The sequence in Figure 5a is confusing by starting with a receiver. The labeling in Figure 5c-d is not clear.

(12) The English writing of the manuscript needs a further polishing. For example, Line 149 "... or nor...", Line 170 "... is a mere...", Line 172 "... is limited to responding..."

Reviewer #3 (Remarks to the Author):

This manuscript reports an innovative result that a single photodetector can only respond within a certain frequency range and can be adjusted by materials manipulation. The simple but effective device design of this work is revolutionary for spatially coupled optical communications, which often suffer from interference. Such complex functions can be realized by a simple single-step spin-coating method without tandem device structure. I support the publication of this work in Nature Communications. To make it more appealing to the readers, the authors may consider the following suggestions:

1. The response time of photodiodes based on perovskite materials is usually around μs . Why can this device achieve an ultra-fast response of less than 20 ns in such a large area (9 mm²)?
2. When controlling the thickness of the hole transport layer and the perovskite layer, the external quantum efficiency (EQE) of almost all devices around 350 nm is much larger than that at other wavelengths. What is the reason for this phenomenon?
3. In the etching UPS test, the Fermi level and valence band position at the perovskite surface are significantly different from the bulk phase. Is this a measurement error or is it due to some unique properties of 2D perovskites?
4. There are many types of hole transport layers, why choose PEDOT:PSS and P3HT as hole transport layers for this work?
5. In the PL test, the emission peak corresponding to 3D perovskite is around 750 nm, while in the TA test, the center position of the emission peak of 3D perovskite is around 725 nm. What may be the reason for the difference in the position of the emission peak?

Reviewer #4 (Remarks to the Author):

In this manuscript, the authors assert that with their perovskite device light-fidelity optical communication is feasible for transmitting text or video data even in the presence of light interference. The feasibility is based on the frequency selective optical response of a single perovskite device. The authors were able to form 2D-3D-2D asymmetric perovskite structures in a one-step process, which is the key factor supporting their claim. However, it is unclear how the phase separation of perovskite differs from previously reported research, and it is difficult to see how its application in optical communications represents a significant advance over current technologies. Nature Communications is a high-quality, interdisciplinary journal that focuses on notable new findings that merit dissemination in their respective fields. Overall, we find the manuscript to be well structured. We enjoyed reading the manuscript, which expanded our understanding of a specific topic. However, we believe it is more appropriate for publication in a specific technology journal than here. To further improve the quality of your paper, please consider the following comments and very

important questions.

1. While we agree with the authors that phase separation can occur in the solution process, we are not convinced that the measured data for asymmetric 2D-3D-2D perovskites in this manuscript are correct. Without the use of an accelerator beam, GIXRD is generally difficult to measure reliably below 0.5 degrees. The peak at 0.1 degree seems anomalous, raising the question of whether 2θ is an accurate value. Is the depth at which the 2D perovskite peak exists similar to the thickness predicted from other data? In how many samples was 2D perovskite formation possible through this phase separation?

2. The presence of the NPN structure is demonstrated in Figure 3 and Figure 4. Given that the perovskite structure might be impacted by the cutting and etching process of the device's cross-section, what methods are used for calibration? Are there any CPD outcomes showing measurements with no transport layer and single-sided absence of electrodes? The potential variation second derivatives in both transport layers appear considerably large in Figures 3g, 3h, leading to uncertainty about the existence of npn and pn alternation before and after light application. The results in Figure 4 pertain to carrier transport and may not warrant inclusion in the text.

3. The optical response of the perovskite device, which is claimed to be frequency selective for code transmission, outperforms the 3D perovskite used in the comparison. Please provide additional data that includes a characterization of the 3D perovskite. The degree of data distortion can be judged differently depending on the criterion of sharpness, and the video signal presented appears to be significantly affected by low-frequency light. In Figure 5, only simple concepts and application data for optical communication are presented, which seems to be a disappointing conclusion in light of the title and keywords of the paper. If the change of electromotive force from npn structure to pn structure in 2D-3D-2D can be induced, the application of in-memory devices to control the accumulation and discharge of charge coupled with optoelectronic logic functions that can be defined under various wavelength and frequency conditions would be a good research direction.

Reviewer #2 (Remarks to the Author):

In this manuscript, the authors report an asymmetric 2D–3D–2D perovskite structure to achieve a frequency-selective photoresponse, which can prohibit light interference up to 454 mW cm^{-2} . The concept that combining two opposite diodes in a single device with different cut-off frequencies to realize frequency-selectivity is very interesting. The manuscript has exhibited comprehensive investigations in designing the device structure and the composition of each functional layers. The demonstration of free-space communication with strong anti-interference performance is impressive. However, several concerns should be addressed in the current version as below.

(1) In the manuscript, the authors use different description of the thin film structure, including “front side” and “back side”, “lower part” and “upper part”, “upper surface”, “frontal”, “glass side”, “perovskite side”, etc., which are suggested to use uniform description and add clear labeling in Figure illustration.

Response: Thank you for your suggestions. In accordance with your guidance, we have simplified the expressions in the manuscript. Based on the characterization descriptions from the different directions in other literature (Nat. Rev. Mater. 2023, 8, 533–551; Adv. Mater. 2020, 33, 2002582; Adv. Mater. 2022, 35, 2207345), we incorporated definitions of the front and back sides in the PL test and added definitions of the lower and upper parts of the perovskite in the caption of Fig. 1b. Furthermore, to more accurately convey the direction of carrier movement, we have indicated our defined positive direction of current in Fig. 1b and standardized the symbol expression of current across all figures in accordance with this.

(2) Please explain why there are negative EQE shown in Figure S5.

Response: Thank you for the comments. Our device structure bears similarity to certain multilayer stacked series devices, with a typical example being the one mentioned in Nat. Commun. 2022, 13, 720. To illustrate the parameters in Fig. 2a, the absorption at 300–350 nm was attributed to PEDOT:PSS. Carrier separation occurred at the PEDOT:PSS-ITO interface, with electrons moving towards ITO and holes moving

towards PEDOT:PSS. The remaining materials could be approximated as resistors, which did not contribute to the photoresponse, resulting in a negative response. Above 350 nm, the absorption of the lower surface of the perovskite began to gradually increase. At this point, the nearest PEDOT:PSS perovskite electric field began to play a role. PEDOT:PSS extracted the photogenerated holes from the perovskite, and these holes were directed to the ITO. The electrons diffused towards the upper interface, resulting in a positive response. As the wavelength further increased, the region where photogenerated carriers were generated further shifted towards the upper interface of the film. The P3HT perovskite electric field began to compete with the PEDOT:PSS perovskite electric field, and the positive photoresponse decreased and eventually became negative again near 600 nm. The devices with other parameters in Supplementary Fig. 7 exhibited a similar trend. However, due to the different thicknesses of the transport layer and absorption layer and different component distributions, the positions of the wavelength transition points were not identical.

(3) The bias applied on the device when performing photodetection test should be mentioned. As described in the Experimental Section, the current was recorded using an oscilloscope. However, the oscilloscope can only record voltage signal. Did any amplifier used in the experiment?

Response: Thank you for the valuable comments. This device, functioning as a self-driven photodetector, does not necessitate an additional bias. The current signal recorded by the oscilloscope is converted with a 50-ohm resistor. To circumvent additional delays induced by amplifier bandwidth issues, amplifiers were not used in the speed tests. However, weak signal detection poses a greater challenge for embedded systems; hence, a preamplifier based on OPA657 was only utilized when signal transmission occurred. Descriptions of related experimental conditions have been incorporated into the experimental section of the manuscript.

(4) Line 150, the authors claim that "... the alteration of the functional layer thickness within a certain range does not directly impact the central frequency." Line 92, another description "the perovskite, PEDOT:PSS, and P3HT thicknesses are critical for achieving frequency-selective photoresponse." Do the two descriptions contradict each

other? The elucidation is not clear.

Response: We appreciate the reviewer's insightful discussion. As depicted in Supplementary Fig. 9, once the selected material is determined, the approximate position of the central frequency (the frequency with the maximum response amplitude in the high-frequency region) is also established; hence, in the original manuscript, it was stated that the thickness of the functional layer did not influence the central frequency. However, if the thickness of each functional layer is not optimized, the device will exhibit leakage in the low-frequency area, compromising the frequency selectivity of the device. Therefore, in the original manuscript, it was stated that the thickness of the functional layer is pivotal to the implementation of a frequency selective response. For instance, in Supplementary Fig. 9b, irrespective of how the thickness of the P3HT layer was altered, the central frequency in the high-frequency region always occurred at 500 kHz. However, if the thickness of the P3HT layer was not finely adjusted to balance the two built-in electric fields, a significant response was observed in the low-frequency region less than 100 Hz, which was entirely detrimental to the implementation of anti-interference. To more accurately convey the above meaning, the above examples were added to the manuscript.

(5) Are Figure 3a and b the AFM topographic images or the KPFM surface potential (SP) images? To demonstrate the surface potential under dark and illumination conditions, KPFM images are more suitable. Dashed lines should be added into KPFM images to show the position corresponding to the SP line profiles in Figure 3c and d.

Response: Thank you for your suggestion. In accordance with your suggestions, we have substituted the AFM morphology map with a surface potential map and used dashed lines to differentiate various regions.

(6) Was the KPFM DC bias applied on the tip or on the sample when performing KPFM? Usually, the DC bias is applied to AFM tip, and consequently the higher Fermi level (lower work function) exhibits higher surface potential. Line 199, the accumulation of electrons in perovskite will lift the Fermi level, as well as the surface potential, which is inconsistent with Figure 3c and d. Please add more experimental details and make explanations.

Response: Thank you for your reminder, which has highlighted the need for a more detailed description of the experimental conditions. We have now included additional information regarding bias and drive routing in the experimental section. During the KPFM test, no extra bias was applied, but the drive routing of the signal recorded during the test was selected on the sample. As you rightly noted, if the tip is chosen as the driving route, the accumulation of electrons would indeed elevate the Fermi level, as well as the surface potential. However, since we opted for the sample as the drive route, the surface potential displayed a decrease, despite electron accumulation. To illustrate this phenomenon, we conducted tests on the surface KPFM of perovskite and highly oriented pyrolytic graphite (HOPG, BRUKER) using both types of drive routing, as depicted in Figure R1. It is well known that the work function of HOPG is 4.6 eV, while that of the perovskite surface is approximately 4.8 eV (Fig. 3j). The difference in surface potential between the two drive routings can be effectively converted into their corresponding Fermi levels.

Figure R1. a, b, c, d, KPFM images of highly oriented pyrolytic graphite (HOPG) (a and b) and the N3 perovskite surface (c and d) at different drive routes. The scale bar is 600 nm. e, f, Detailed potential curves of tip drive routing (e) and sample drive routing (f).

(7) The authors mentioned that the perovskite contains 2D-3D-2D phase separation. But why do the UPS results (Figure 3i-k) not show the change of Fermi level and

valance band of perovskite at PEDOT:PSS interface similar to P3HT-Perovskite interface?

Response: Thank you for your comments. KPFM provides a characterization of the entire device, whereas UPS only represents the absorption layer. UPS characterization through etching can accurately depict the details of the potential changes in the depth direction of the perovskite thin film. However, this does not reflect the potential of the final device, as the interface will undergo changes due to the influence of the two layers of the transport materials. The cross-sectional analysis of KPFM can better represent the potential change trend of the final device. However, due to the limitations of the lateral resolution of probe microscopy technology, abrupt changes in potential will be reduced. A direct manifestation of this discrepancy is that according to the UPS results, the Fermi level at the upper surface of perovskite is quite high, with a difference from the bulk reaching hundreds of meV, whereas the KPFM results show a difference of only tens of meV. Moreover, in an effort to mitigate the potential damage to the sample caused by high-energy sectioning techniques such as the focused ion beam (FIB), we used the sample preparation procedure delineated in the referenced literature (Science 2020, 367, 1135-1140; Nat. Energy 2019, 4, 150–159; Nature 2023, DOI: 10.1038/s41586-023-06784-0). However, this approach does not ensure a cross-section that is strictly perpendicular to the surface. In instances where a certain tilt angle is present, the perceived layer thickness may be exaggerated. As a result, we derived Figure R2 by proportionally adjusting the horizontal axis of the UPS and KPFM results. As illustrated in Figure R2 (for clarity, the vertical axis has been transposed to the work function), a pattern similar to that observed in UPS was still discernible in the KPFM results. The observed increase in the work function on either side of KPFM relative to UPS could be ascribed to the effects of P3HT and PEDOT:PSS. As an additional note, the abrupt high Fermi level change on the upper surface of perovskite was due to the deposition of excess 2D salt (Adv. Energy Mater. 2020, 10, 2000687; Adv. Mater. 2021, 33, 2101714).

Figure R2. Work function and CPD diagram for N3 perovskite film at different depths.

(8) The device used in free-space optical communications is ITO/SnO₂/3D perovskite/Spiro-OMeTAD/Ag, which is completely different from the one discussed above, i.e., Ag/P3HT/perovskite/PEDOT:PSS/ITO device. It is very strange to use a device never discussed above for demonstration.

Response: Thank you for your advice. Our goal was to illustrate that the anti-interference functionality is unattainable in conventional n-i(3D)-p structured devices. As such, we used the ITO/SnO₂/3D perovskite/Spiro-OMeTAD/Ag device configuration as outlined in the manuscript, which is widely used as a control group for perovskite detectors (Nature 2020, 583, 790–795; Adv. Mater. 2022, 34, 2206957; Adv. Mater. 2023, 35, 2210068). Nonetheless, a solitary comparison does have certain inadequacies. Therefore, in response to your suggestion, we incorporated a comparative analysis of the Ag/P3HT/perovskite/PEDOT:PSS/ITO structure (p-i(3D)-p) (Supplementary Video 3) and the Ag/PCBM/perovskite/PEDOT:PSS/ITO (p-i(2D-3D-2D)-n) structure (Supplementary Video 4). Given that these structured devices all exhibit substantial leakage current at lower frequencies, the attainment of frequency selectivity is precluded. Nevertheless, certain disparities can be discerned. Conventional n-i(3D)-p devices invariably lack any frequency selectivity. Utilizing an interference source of 55.1 mW cm⁻² intensity, lethal interference can still be induced for character transmission even at the maximum distance. The p-i(2D-3D-2D)-n device also lacks any frequency selectivity. In the absence of a blocking layer, the perovskite's

V-shaped band structure is incapable of charge accumulation. The p-i(3D)-p device can barely withstand the interference of 55.1 mW cm^{-2} background light. In the presence of blocking layers on both sides, the perovskite can be approximated as the base of the V-shaped band; however, at this juncture, the relative relationship between the band of the CTL and the perovskite has been established, lacking a variable to balance the two intrinsic electric fields. The ultimate performance of the device is shown in the device depicted in Supplementary Fig. 9; this device exhibits a significant leakage current at lower frequencies. Consequently, effective frequency selectivity remains unachievable.

(9) In the video file (445007_0_video_61698_rwpbf1) that shows the data transmission of “Soochow University”, when the light interference illuminates the receiver, the baseline of signals in oscilloscope apparently lifts, indicating that the device responds to the light interference. This is different from the mechanism that the responses of opposite diodes can cancel each other at low frequency. The frequency-dependent responsivity of photodetector should be included and discussed.

Response: Thank you for your comments. The device’s frequency-related response is depicted in Fig. 2f. The monochromatic light response (450 nm) of the device at low frequencies ranges from approximately 3% to 16% of the central frequency. Under the interference of broadband sunlight, the device’s suppression capability is enhanced due to the cancellation of positive and negative currents, but it currently cannot reach absolute zero. Therefore, when the interference light shines on the photodetector, there is a baseline lift. However, taking the low-frequency flicker interference as an example, the signal light intensity we use is 5.84 mW cm^{-2} (at the logic high state), and the interference light intensity is 227 mW cm^{-2} (average power, 50% duty ratio). The frequency selective device exhibits a limit lift that is less than 20% of the total amplitude, whereas the baseline lift of traditional detection devices is escalated to 100% of the original amplitude (saturated). The comparison between the two reflects the frequency selectivity of the device. Higher frequency suppression ratios can be achieved through further regulation of CTL, as shown in Supplementary Fig. 9. For a specific wavelength, it is possible to achieve an absolute zero response at low

frequencies. However, limited by the different positions of photogenerated carriers excited by different wavelengths, to achieve an absolute zero response at low frequencies for any wavelength, new absorption layer and transport layer design strategies may be needed; these can include the following: 1. Through optical design, the absorption coefficient of materials can be further enhanced to localize the region where the photogenerated carriers are generated, which can avoid imbalance problems caused by different positions where carriers are generated. 2. Through more built-in electric field designs in the layers, two frequency-selective monomers with the same central frequency are connected in series to further enhance the low-frequency suppression capacity.

(10) Because the letters data are encoded by ASCII as digital signals, is there a comparator used for determining high (1) and low (0) levels? Is it suitable to use digital signal to prove the anti-interference performance? Besides, more experimental details should be added, including the laser intensity irradiated on photodetector and the encoding methods for video transmission.

Response: Thank you for the comments. In the character transmission demonstration, comparators are set at 20%/80% V_{pp} positions and are used to determine low/high levels. In the current field of optical communication, digital signals have always been mainstream signal carriers (Nat. Electron. 2020, 3, 156-164; Nat. Commun. 2021, 12, 1666; Nat. Commun. 2021, 12, 5076; Adv. Funct. Mater. 2022, 32, 2208694). However, using only digital signal transmission to demonstrate anti-interference characteristics is not complete because the setting of the comparator will affect the specific values of anti-light noise. Nevertheless, setting them at the same 20%/80% V_{pp} position enables a comparison of the noise resistance capabilities from the different samples. Moreover, an analogue video signal based on the phase alternating line (PAL) system is also used to demonstrate the anti-interference capability (Supplementary Video 5, 6, and 7). Therefore, both digital and analogue signal transmission noise resistance capabilities have been proven. The current shortcoming is that due to the bandwidth of traditional perovskite photodetectors being lower than 1 MHz, it cannot meet the transmission requirements of PAL analogue video. We have added a video transmission comparison

based on silicon detectors (model: S6968, Hamamatsu) to enhance the contrast in Supplementary Video 7.

In addition, the light intensity shining on the photodetector is 5.84 mW cm^{-2} , and the video transmission method modulates the camera data into PAL format, which is used as the driving current of the laser diode for intensity modulation after amplification. After the detector collects the current signal, it is input to the acquisition card and converted into a video streaming output to the screen. The above details have been added to the caption of Fig. 4.

(11) The sequence in Figure 5a is confusing by starting with a receiver. The labeling in Figure 5c-d is not clear.

Response: Thank you for your advice. Your suggestion is indeed more reasonable, and the corresponding adjustments have been made in the figure. The unclear labelling in Fig. 4c-d appears to be a result of the review system's compression. To address this, high-definition images have been uploaded as separate attachments this time.

Figure R3. High-definition images of Fig. 4c and d.

(12) The English writing of the manuscript needs a further polishing. For example, Line

149 "... or nor...", Line 170 "... is a mere...", Line 172 "... is limited to responding..."

Response: We apologize for the impediment to your reading process, due to the nonstandard grammatical construction of the article. Our manuscript has been polished by AJE under Nature before submission, and we have contacted them for further modifications.

Figure R4. Recertification report of the manuscript polished by AJE.

Reviewer #3 (Remarks to the Author):

This manuscript reports an innovative result that a single photodetector can only respond within a certain frequency range and can be adjusted by materials manipulation. The simple but effective device design of this work is revolutionary for spatially coupled optical communications, which often suffer from interference. Such complex functions can be realized by a simple single-step spin-coating method without a tandem device structure. I support the publication of this work in Nature Communications. To make it more appealing to the readers, the authors may consider the following suggestions:

1. The response time of photodiodes based on perovskite materials is usually around μs . Why can this device achieve an ultra-fast response of less than 20 ns in such a large area (9 mm^2)?

Response: Thank you for your comment. The response speed of photodetectors is indeed limited by the carrier mobility of the material and the parasitic capacitance of the device (Nat. Photon. 2022, 16, 718–723; Nat. Photon. 2020, 14, 578–584). However, importantly, the capacitance accumulates at each interface. The multijunction stacked device structure designed in this study essentially constitutes a series of capacitances, thereby reducing the total capacitance value and consequently reducing the delay caused by capacitance (Adv. Mater. 2020, 32, 1908108). Additionally, from the perspective of carrier transport, there is a more intuitive understanding. From the KPFM test, at the moment of dark-light conversion, as shown in Figure R1, the photogenerated electrons will accumulate towards the middle of the bulk phase, and photogenerated holes will simultaneously separate at both interfaces. The PEDOT:PSS side is closer to the position where the photogenerated carriers are generated and has a stronger electric field. The photogenerated holes will reach this interface faster and generate a response in the external circuit. This process dominates the generation of the device response rising edge. The photogenerated holes on the P3HT side will subsequently reach the upper interface, offsetting the original response and thereby promoting the start of the falling edge. This near-interface “active generation-active cancellation” response

mechanism allows the device to eliminate the original “active generation-external circuit passive recombination” process where the carrier transport distance is long and the recombination speed is slow, further improving the response speed. In fact, our design is expected to break through the constraints of the RC parameters on the traditional optical detector bandwidth, making large-area high-speed optical detection possible. Since the main point of this article is not to explain the improvement of response speed, we added a brief explanation of this response process in the manuscript.

Figure R1. Schematic of carrier distribution under dark and light conditions at high-frequency switching.

2. When controlling the thickness of the hole transport layer and the perovskite layer, the external quantum efficiency (EQE) of almost all devices around 350 nm is much larger than that at other wavelengths. What is the reason for this phenomenon?

Response: Thank you for your comment. Regardless of the changes in the thickness of the CTL and perovskite layers, large carriers are transported in the same direction at approximately 350 nm (Supplementary Fig. 9). This is attributed to the fact that light with a wavelength less than 350 nm will be absorbed by PEDOT:PSS and generate a photocurrent. In this scenario, the perovskite and P3HT layers mainly act as resistors; hence, the current at 350 nm will not be suppressed to zero, and the direction will not change.

3. In the etching UPS test, the Fermi level and valence band position at the perovskite surface are significantly different from the bulk phase. Is this a measurement error or is it due to some unique properties of 2D perovskites?

Response: Thank you for your comments. The significantly different Fermi energy level positions of the 2D perovskite surface area and the bulk phase were not due to

testing errors. This occurred because the 2D perovskite thin film prepared by the hot-casting method was prone to form an excess of 2D salt enrichment on the surface. In the SEM image in Supplementary Fig. 17a, many alternating bright and dark positions on the surface were found to be the enriched 2D salt area and the perovskite area. This surface 2D salt enrichment state was also found in our previous work (Adv. Mater. 2021, 33, 2101714) and others' work (Adv. Energy Mater. 2020, 10, 2000687). Due to this characteristic, the 2D salt on the surface was easily enriched, and a 2D perovskite layer formed on the upper surface, thereby spontaneously forming a 2D-3D-2D perovskite distribution.

4. There are many types of hole transport layers, why choose PEDOT:PSS and P3HT as hole transport layers for this work?

Response: Thank you for your comment. Indeed, PEDOT:PSS and P3HT are commonly used hole transport layer materials due to their high hole mobility, which contributes to a fast response speed. Due to their excellent properties, they are a popular choice in the design of optoelectronic devices.

5. In the PL test, the emission peak corresponding to 3D perovskite is around 750 nm, while in the TA test, the center position of the emission peak of 3D perovskite is around 725 nm. What may be the reason for the difference in the position of the emission peak?

Response: Thank you for your comment. In the characterization of the 2D perovskite system, we observed that this was a relatively common phenomenon, and similar phenomena were found in other studies (Adv. Funct. Mater. 2021, 31, 2008404; J. Phys. Chem. Lett. 2020, 11, 1120–1127; Nano-Micro Lett. 2023, 15:91). These two different emission peaks originated from the different mechanisms of these two characterizations. The PL recorded the fluorescence signal generated by radiative recombination. Since the energy generated by 2D perovskite spontaneously transferred to 3D perovskite, the main body of radiative recombination was 3D perovskite. Therefore, the position of this emission peak corresponded to the energy difference between the valence band and conduction band of the 3D perovskite. The principle of transient absorption (TA) is to first excite the semiconductor to an excited state, and then subsequent light passing through the semiconductor will produce different absorption values compared to the

ground state. The TA spectrum was obtained based on the difference in absorption values; thus, this value was the absorption difference between the excited state semiconductor and the intrinsic state semiconductor, and the characterized substance underwent some changes. Moreover, the exciton binding energy of 2D perovskite was significantly enhanced compared to that of 3D perovskite. Therefore, when exciting a specific bandgap semiconductor, energy higher than the bandgap width was needed; therefore, in TA, the ground-state bleaching peak shifted towards high energy compared to the PL emission peak.

Reviewer #4 (Remarks to the Author):

In this manuscript, the authors assert that with their perovskite device light-fidelity optical communication is feasible for transmitting text or video data even in the presence of light interference. The feasibility is based on the frequency-selective optical response of a single perovskite device. The authors were able to form 2D-3D-2D asymmetric perovskite structures in a one-step process, which is the key factor supporting their claim. However, it is unclear how the phase separation of perovskite differs from previously reported research, and it is difficult to see how its application in optical communications represents a significant advance over current technologies. Nature Communications is a high-quality, interdisciplinary journal that focuses on notable new findings that merit dissemination in their respective fields. Overall, we find the manuscript to be well structured. We enjoyed reading the manuscript, which expanded our understanding of a specific topic. However, we believe it is more appropriate for publication in a specific technology journal than here. To further improve the quality of your paper, please consider the following comments and very important questions.

Response: We appreciate the recognition from the reviewers for our manuscript. We believe that our manuscript aligns well with high-quality comprehensive journals such as Nature, Science, and Nature Communications. This is because our paper not only encompasses the technical implementation of a singular frequency-selective response detector but also delves into the elucidation of the mechanism - the design of the built-in electric field in multijunction devices. The designed device also delivers a plethora of new phenomena that are not discussed in detail in the article; these include wavelength-modulated bipolar responses that hold potential for application in neuromorphic devices, wavelength selectivity that can be utilized for colour discrimination detection, and a novel response mechanism that potentially surpasses traditional RC time limitations. Consequently, this manuscript is poised to engage readers interested in the interdisciplinary fields of physics, materials, and electronics. This aligns seamlessly with the ethos of the Nature Communications journal, a

multidisciplinary platform committed to publishing high-quality research spanning all areas of the biological, health, physical, chemical, earth, social, mathematical, applied, and engineering sciences.

In response to your statement of “it is unclear how the phase separation of perovskite differs from previously reported research,” we would like to provide further clarification. The phenomenon of phase separation in perovskites has been extensively reported by numerous researchers. However, due to the requirements of the photovoltaic application field, these reports primarily focus on eliminating phase separation to enhance photovoltaic conversion efficiency rather than utilizing it to achieve specific functions (Nat. Energy 2021, 6, 38–45; Nat. Rev. Mater. 2023, 8, 533–551). In 2021, we employed hot-casting combined with DMSO to induce controlled phase separation, thereby achieving a gradient energy level distribution and significantly improving the photoelectric detection performance (Adv. Mater. 2021, 33, 2101714). With controlled phase separation, we can achieve various vertical structure devices, including the frequency-selective detector with a V-shaped built-in electric field proposed in this work. Regarding the controllability of this phase separation, we added a substantial number of grazing incidence characterizations to demonstrate its universality and controllability. However, although hot-casting-induced perovskite-controlled phase separation technology is used in this work, the device designed in this paper does not necessarily depend on it. The pivotal mechanism of the frequency selectivity achieved in our study has been proven to be the design of the longitudinal distribution of built-in electric fields in devices, which is not limited to a single material system. The realization through perovskite phase separation control technology is just an example because this all-solution one-step construction process is simple and inexpensive, and we expect that this process will be further developed. By transforming silicon-based phototransistors (model: 3DU33 and ST-1KLB, NPN type), as shown in Figure R1, similar frequency-selective responses can still be achieved. Nonetheless, the capacity for manipulation of silicon materials through the chemical solution method remains profoundly constrained, and the attainment of a reduced leakage current at lower frequencies coupled with an enhanced response speed continues to be elusive.

Figure R1. a,b, Normalized response of the NPN phototransistor varying with the frequency for Model: 3DU33 (a) and Model: ST-1KL3A (b).

In response to your statement of “it is difficult to see how its application in optical communications represents a significant advance over current technologies,” I would like to further explain the application potential of this work in future optical communications: The interference problem is one of the most severe challenges hindering free-space coupled optical communication; this fact has been acknowledged by numerous researchers, and solutions have been sought (Nat. Photon. 2023, 17, 814–821; Nat. Photon. 2023, 17, 791–797; Nat. Commun. 2020, 11, 1171; Nat. Commun. 2020, 11, 6368; Nat. Commun. 2022, 13, 1476; Light Sci. Appl. 2019, 8, 27).

However, in the face of broad-spectrum, high-intensity natural light interference, there exists no method capable of precluding interference signals from infiltrating the communication system. For the first time, in our study, a frequency selection method from electrical parameter characteristics is proposed to address this challenge and provides a general design mechanism. For AM1.5 broadband sunlight interference, frequency selective detectors can provide an additional order of magnitude noise suppression for the system. Moreover, this approach does not conflict with existing paths through optical filtering, indicating that an enhanced level of interference suppression could be realized through their integration. For space-coupled optical communication, this implies lower transmission power, longer effective communication distance, and of course, lower cost due to perovskite-based all-solution

one-step phase separation control. I believe that these are important objectives for any optical communication system, and I hope you will agree with this deduction.

Thank you once again for your valuable comments, and we hope our response resolves your concerns regarding the suitability of the journal and the originality of the results.

1. While we agree with the authors that phase separation can occur in the solution process, we are not convinced that the measured data for asymmetric 2D-3D-2D perovskites in this manuscript are correct. Without the use of an accelerator beam, GIXRD is generally difficult to measure reliably below 0.5 degrees. The peak at 0.1 degree seems anomalous, raising the question of whether 2θ is an accurate value. Is the depth at which the 2D perovskite peak exists similar to the thickness predicted from other data? In how many samples was 2D perovskite formation possible through this phase separation?

Response: Thank you for your constructive opinions. In response to your query regarding the reliability of GIXRD characterization, we have found in other literature that GIXRD can be used to conduct tests down to 0.5° and even 0.1° (Nat. Energy 2023, 8, 989–1001; Nat. Photon. 2023, 17, 856–864; Nature 2023, DOI: 10.1038/s41586-023-06784-0). These characterizations are often used to illustrate the existence of different phases and stresses between the surface and the bulk, and the splitting of the peaks indicates the presence of different phases. Our XRD instrument is equipped with a LYNXEYE XE-T detector, which has an energy resolution better than 380 eV at 8 keV ($\lambda = 0.154$ nm); thus, it can display signals at low grazing angles. To further substantiate this deduction, we used a synchrotron radiation source for grazing incidence wide angle X-ray scattering (GIWAXS) testing at beamlines BL03HB at the Shanghai Synchrotron Radiation Facility (SSRF) in Supplementary Figs. 4 and 5. We conducted tests at grazing angles of 0.1° and 0.5° and found that peaks corresponding to 2D perovskite appeared at angles corresponding to GIXRD, confirming the existence of 2D perovskite on the upper surface. A comprehensive comparison is presented in Figure R2. To facilitate comparison, we transformed the 2θ values from GIXRD into q values. The XRD peak, situated at approximately 10° and corresponding to $q = 0.7 \text{ \AA}^{-1}$, was notably

absent, attributed to the gap present in the detector (model: PILATUS 2 M). Intriguingly, due to the enhanced resolution of GIWAXS, a new 2D perovskite Debye-Scherrer diffraction ring was identified at $q = 0.22 \text{ \AA}^{-1}$ under a grazing angle of 0.1° (Supplementary Fig. 4a and 5a), further corroborating the presence of surface 2D perovskites. The 2D diffraction peaks near the (111) and (202) planes ($q \approx 1$ and 2 \AA^{-1}) were derived from 2D perovskites with large n values (Nat. Mater. 2018, 17, 900–907).

In response to the question “Is the depth at which the 2D perovskite peak exists similar to the thickness predicted from other data?,” this form of 2D perovskite exists in a mixed state with 3D perovskite, not strictly layered; thus, it is impossible to determine exactly how thick the 2D perovskite film is. However, through calculation, we can obtain the X-ray penetration depth at different grazing angles in Figure R3 (Adv. Funct. Mater. 2021, 31, 2103252). In Supplementary Fig. 2, we found that the appearance, disappearance and half-peak width change of the peaks were always present when changing the grazing angle; therefore, the distribution of specific phase components could not be predicted. However, some information could be corroborated from the side. For example, in the UPS test, an evident Fermi level that was different from the bulk phase in the area less than 100 nm was observed; additionally, in the GIXRD test, at a grazing angle of 0.6° , the 2θ crystal face at approximately 13.5° disappeared, and the X-ray penetration depth at this time was approximately 78.3 nm. In the GIWAXS test, as the grazing incidence angle increased, the peak intensity of the 2D perovskite showed a negligible change, while the peak intensity of the 3D perovskite showed a sharp increasing trend; this result indicated that the 2D perovskite was enriched on the surface, and the 3D perovskite was significantly increased in the bulk phase (Energy Environ. Sci. 2022, 15, 3369-3378).

In order to response the question “In how many samples was 2D perovskite formation possible through this phase separation”, fifteen samples of N3 perovskite data based on tBBAI have been provided in Figure R4, R5, and R6 to demonstrate the repeatability of our experiment. Due to the non-horizontal nature of the substrate, diffraction spectra among samples deviate from perfect congruence owing to errors in grazing incidence angles (mainly occurs at grazing angles less than 0.5°). Nevertheless, the diffraction

peaks at 13.6° and 27.5° , characteristic of 2D perovskite, diminish and disappear at high grazing angles across all samples. Simultaneously, peaks indicative of 3D perovskite, around 14.2° and 28.5° , gradually emerge and dominate. These findings collectively affirm the enrichment of two-dimensional perovskite on the surfaces of all samples, proving the repeatability of the experiment. In order to prove the universality of this method, we also tested commonly used 2D salts: PEAI, and BAI in Supplementary Figs. 13, and 14, respectively. We found that there are diffraction peaks of 2D perovskite on the surface at small grazing angles, proving the existence of 2D perovskite on the upper surface.

Figure R2. a, b, c, GIXRD patterns transferred from Supplementary Figure 2. d, e, f, Polar intensity profiles averaged along with the rings calculated from GIWAXS by Fit2D software.

Figure R3. X-ray penetration depth as a function of the grazing angle. The perovskite material is selected as MAPbI_3 .

Figure R4. Enlarged GIXRD patterns with a grazing angle from 0.1° to 5° of the sample 1 to sample 5.

Figure R5. Enlarged GIXRD patterns with a grazing angle from 0.1° to 5° of the sample 6 to sample 10.

Figure R6. Enlarged GIXRD patterns with a grazing angle from 0.1° to 5° of the sample 11 to sample 15.

2. The presence of the NPN structure is demonstrated in Figure 3 and Figure 4. Given that the perovskite structure might be impacted by the cutting and etching process of the device's cross-section, what methods are used for calibration? Are there any CPD outcomes showing measurements with no transport layer and single-sided absence of electrodes? The potential variation second derivatives in both transport layers appear considerably large in Figures 3g, 3h, leading to uncertainty about the existence of npn and pn alternation before and after light application. The results in Figure 4 pertain to carrier transport and may not warrant inclusion in the text.

Response: We apologize that the testing conditions for interface KPFM were not clearly explained in the experimental section, as we used a sample rather than a tip drive routing method. Therefore, the perovskite thin film should exhibit a PNP structure. The sample preparation methods followed those outlined in these articles (Science 2020, 367, 1135-1140; Nat. Energy 2019, 4, 150–159; Nature 2023, DOI: 10.1038/s41586-023-06784-0). Notably, the focused ion beam (FIB) sampling was circumvented to avert potential damage to the cross-sectional samples. The Fermi energy levels of the surface and bulk phases provided by UPS were calibrated using Au and highly oriented pyrolytic graphite (HOPG, BRUKER), respectively (Figure R5). The Fermi energy level converted by KPFM aligned well with the Fermi energy level given by UPS, indicating that the cross-sectional samples were not damaged.

In response to the question “Are there any CPD outcomes showing measurements with no transport layer and single-sided absence of electrodes?,” the comprehensive device configuration we used is Ag/P3HT/perovskite/PEDOT:PSS/ITO. Consequently, the term “no transport layer and single-sided absence of electrode” pertains to a device structure consisting solely of “perovskite-ITO.” The outcomes are depicted in Supplementary Fig. 20, where the unadulterated perovskite film in isolation can have a V-shaped CPD distribution. Simultaneously, devices incorporating a single-sided CTL, as delineated in Supplementary Figs. 26 and 27, persist in maintaining a V-shaped CPD structure, with the P3HT layer serving a pivotal function in the accumulation of photogenerated carriers. In addition, we have ascertained that the utilization of a Cu top electrode does not exert an influence on the potential fluctuation of the ultimate device,

as shown in Figure R6. They display a pattern similar to that with Cu electrodes, indicating that the electrode may induce a minor modification to the CPD but will not impinge on the final V-shape.

In response to your statement of “The potential variation second derivatives in both transport layers appear considerably large in Figures 3g, 3h, leading to uncertainty about the existence of npn and pn alternation before and after light application,” upon illumination, the potential difference at the interface between P3HT and perovskite experiences a significant reduction, a phenomenon attributable to the accumulation of the photogenerated electrons. As a result, it can be postulated that with the provision of additional carriers, this structure may transition from a PNP to an NP configuration. Nevertheless, during the KPFM examination under illuminated conditions, as depicted in Figure R7, the intensity increases to a large value of 554.2 mW cm^{-2} (due to the limitation of the spectrometer measurement range, the actual light intensity should be higher). This implies that while such a transition is feasible, an exceedingly high photon count is needed. Notably, during this examination, we utilized a metallic base for cooling in an endeavour to maintain optimal temperature stability. However, when confronted with a higher intensity of incident interference light (such as on the sunlit side of certain satellites), the predicament of elevated temperature is likely to manifest prior to the onset of light saturation; although, this is not the primary concern of this investigation.

In response to the statement of “The results in Figure 4 pertain to carrier transport and may not warrant inclusion in the text,” here, we wanted to analyse the effect of a single field in detail. In accordance with your suggestion, the content in Fig. 4 has been relocated to the Supplementary Information for further discussion.

Figure R7. a,b,c,d, KPFM images of Au (a), highly oriented pyrolytic graphite (HOPG) (b), cross-sectional N3 perovskite (c), and N3 perovskite surface (d). The scale bar is 500 nm. e,f, Detailed potential curves of N3 perovskite with Au (e) and HOPG (f).

Figure R8. a,b,c, Cross-sectional AFM image (a) of the P3HT/N3 perovskite/PEDOT:PSS/ITO device, corresponding KPFM (b) and measured CPD (c). The green, grey and yellow regions represent P3HT, perovskite and PEDOT:PSS/ITO, respectively. The scale bar is 500 nm.

Figure R9. Light source spectrum for light-state KPFM testing.

3. The optical response of the perovskite device, which is claimed to be frequency selective for code transmission, outperforms the 3D perovskite used in the comparison. Please provide additional data that includes a characterization of the 3D perovskite. The degree of data distortion can be judged differently depending on the criterion of sharpness, and the video signal presented appears to be significantly affected by low-frequency light. In Figure 5, only simple concepts and application data for optical communication are presented, which seems to be a disappointing conclusion in light of the title and keywords of the paper. If the change of electromotive force from npn structure to pn structure in 2D-3D-2D can be induced, the application of in-memory devices to control the accumulation and discharge of charge coupled with optoelectronic logic functions that can be defined under various wavelength and frequency conditions would be a good research direction.

Response: Thank you for your comments. The 3D perovskite utilized in this study is MAPbI₃. In Supplementary Fig. 29, we added the XRD and PL tests for MAPbI₃, demonstrating its excellent crystalline quality. Its morphology and frequency response are depicted in Supplementary Fig. S11a and S15.

In response to your statement of “The degree of data distortion can be judged differently depending on the criterion of sharpness, and the video signal presented appears to be significantly affected by low-frequency light,” we used a video system

based on a phase alternating line (PAL), which is an analogue signal with a fixed resolution and frame rate. The distortion cannot be quantitatively determined by the bit error rate as digital signals can. Notably, we utilized a flashing light (square wave) as the light interference source in the original video transmission display. Intrinsically, in the Fourier transformation of the square wave, the harmonic coefficients are not 0; thus, the square wave is not a waveform of a single frequency. As illustrated in Supplementary Fig. 31, the square wave signal source comprises numerous high-frequency components reaching up to the MHz range. These high-frequency signals are subsequently captured by detectors, leading to interference in the video transmission process. As evidence, when pure low-frequency sinusoidal light is used as the interference signal, no disruption to the signal occurs (Supplementary Video 6).

In response to the question “In Figure 5, only simple concepts and application data for optical communication are presented, which seems to be a disappointing conclusion in light of the title and keywords of the paper,” in Fig. 5, our aim was to demonstrate that the device could perform high-speed data transmission and still work stably under strong light interference; thus, the focus of this figure was to demonstrate the feasibility of practical applications. At present, the figure only shows the basic workflow and some screenshots of the video of stable data transmission because video information cannot be displayed in the main text. To enhance the data presented in this figure, we added supplementary video demonstrations, which include: 1. the Ag/P3HT/perovskite/PEDOT:PSS/ITO structure (p-i(3D)-p) (Supplementary Video 3); 2. the Ag/PCBM/perovskite/PEDOT:PSS/ITO (p-i(2D-3D-2D)-n) (Supplementary Video 4); both are used for character signal transmission. However, their resistance to interference leaves room for improvement, highlighting the necessity of constructing double HTLs and 2D-3D-2D structures. 3. Beyond the lossless transmission of our final device under sinusoidal interference (Supplementary Video 6), we have also incorporated video transmission of commercial silicon detectors (Supplementary Video 7). While silicon detectors possess the capability to swiftly transmit video signals, they succumb to signal loss immediately under minor light intensity interference. This further highlights the significance of our anti-interference capability.

In response to the suggestion “If the change of electromotive force from npn structure to pn structure in 2D-3D-2D can be induced, the application of in-memory devices to control the accumulation and discharge of charge coupled with optoelectronic logic functions that can be defined under various wavelength and frequency conditions would be a good research direction,” we first would like to thank you for your suggestion for further expansion. We have recently been working on related neural network work, and your mention of changing the potential from PNP to NP structure to achieve in-memory devices is very interesting. Combined with the structural principles of MOSFET, we will consider your suggestion in future research endeavours.

REVIEWER COMMENTS

Reviewer #2 (Remarks to the Author):

The authors have done great job in revising the previous version of the manuscript based on the last reviewing report. The clarity of the manuscript is significantly improved. To the current version, I still have several concerns need to be addressed as follows.

- (1) Regarding to the previous Comment 3, the authors responded that the current signal was directly converted by using a 50-ohm resistor and recorded by the oscilloscope. However, by considering the μA -photoresponse of the device, 50-ohm resistor is insufficient to convert it into a significant voltage signal able to be recorded by the oscilloscope. Therefore, I consider that the measurement was performed with a high input impedance on the oscilloscope, i.e., 1M-ohm. Please make a confirmation.
- (2) In Figure 4b, are the waves the real received signals by the detector recorded by the oscilloscope or the transmitted signals on the laser? Please clarify it in the figure caption.
- (3) The expression “juncture” is not a common description of the device structure, which is preferred to use “junction structure” instead.

Reviewer #3 (Remarks to the Author):

The manuscript has been comprehensively revised according to the reviewers' comments. I think it can be accepted without further revision.

Reviewer #4 (Remarks to the Author):

The authors have done a commendable job addressing the issues we raised, providing detailed explanations for the most part, and we acknowledge their effort. While there are some minor gaps in understanding, they do not significantly impact the overall conclusions.

Reviewer #2 (Remarks to the Author):

The authors have done great job in revising the previous version of the manuscript based on the last reviewing report. The clarity of the manuscript is significantly improved. To the current version, I still have several concerns need to be addressed as follows.

Thank you very much for the valuable suggestions you provided during the revision process. In response to your queries, we have provided a detailed explanation for each point. We believe this will alleviate any remaining concerns you may have and further enhance the quality of this manuscript.

(1) Regarding to the previous Comment 3, the authors responded that the current signal was directly converted by using a 50-ohm resistor and recorded by the oscilloscope. However, by considering the μA -photoresponse of the device, 50-ohm resistor is insufficient to convert it into a significant voltage signal able to be recorded by the oscilloscope. Therefore, I consider that the measurement was performed with a high input impedance on the oscilloscope, i.e., 1M-ohm. Please make a confirmation.

Response: Thanks for your valuable comments. The input impedance of our testing system is actually 50-ohm, not 1M-ohm. Typically, oscilloscopes are incapable of detecting current signals, necessitating the use of a current-sense resistor. As depicted in Figure R1, we present the current signal that has been converted by a 50-ohm resistor. In addition, we display the corresponding original voltage signal, which was recorded by a 1M-ohm input impedance. For better visualization, the voltage signal has been reversed. We can see that these two signals are identical.

The choice of current-sense resistor is crucial, and a larger resistor value generates a stronger voltage signal. However, it also causes the device to deviate from the short-circuit operating state, leading to a reduction in operating bandwidth. Given the communication field demonstrated in this article, sacrificing communication bandwidth is unacceptable. Therefore, a smaller resistor is a preferable choice.

Figure R1. a, Current signal converted by a 50-ohm resistor and the corresponding voltage signal. **b-h**, Current amplitude measurement after 0 dB (b), -5 dB (c), -10 dB (d), -15 dB (e), -20 dB (f), -25 dB (g), and -30 dB (h) attenuation.

It's worth noting that the voltage sensing capabilities of oscilloscopes are often underestimated. Even the most common oscilloscope with an 8-bit vertical resolution, set at 1 mV/div, possesses a minimum resolution of approximately 4 μ V. For a 50-ohm current-sense resistor, this translates to a current resolution of around 80 nA. Thus, it's not the oscilloscope resolution that impedes small signal detection, but rather the noise. If the noise issue can be addressed, the current detection capabilities of oscilloscopes could be significantly enhanced. Firstly, electromagnetic shielding must be implemented. All connections should be minimized in terms of length and quantity, and

standard SMA or BNC interfaces should be used (as shown in Fig. 4c and 4d). The oscilloscope's acquisition mode should be set to "high resolution" to improve vertical resolution (promoted to 11 bit, for MDO3102). Lastly, the sync signal of the light source should be used as the trigger source (rather than the signal itself). As demonstrated in Figure R1b to R1h, after implementing the above measures, the signal-to-noise ratio is sufficient to resolve a signal with an amplitude of only μA . The data from Figure R1b to R1h is presented in Figure R2. It can be observed that the background noise is suppressed to a level of less than 300 nA, and signals with μA amplitudes can be measured. To further mitigate the impact of noise, the response amplitude was defined as the average current value under the light state within one cycle. The above-mentioned testing considerations have been incorporated into the experimental section.

Figure R2. The current curve after different attenuation from Figure R1.

(2) In Figure 4b, are the waves the real received signals by the detector recorded by the oscilloscope or the transmitted signals on the laser? Please clarify it in the figure caption.

Response: Thanks for your comment. The signal depicted in Figure 4b represents the transmitted signals from the laser, serving as a contrast to the detector's signal showcased in the video. This point has been added in the figure's caption.

(3) The expression “juncture” is not a common description of the device structure, which is preferred to use “junction structure” instead.

Response: Thanks for your advice. The manuscript has been updated with revised terminology.

Reviewer #3 (Remarks to the Author):

The manuscript has been comprehensively revised according to the reviewers' comments. I think it can be accepted without further revision.

Response: We sincerely appreciate the reviewer's time and effort in reviewing our manuscript. We are grateful for their recommendation to publish our research.

Reviewer #4 (Remarks to the Author):

The authors have done a commendable job addressing the issues we raised, providing detailed explanations for the most part, and we acknowledge their effort. While there are some minor gaps in understanding, they do not significantly impact the overall conclusions.

Response: We sincerely appreciate the reviewer's time and effort in reviewing our manuscript. We are grateful for their recognition to our research.

REVIEWERS' COMMENTS

Reviewer #2 (Remarks to the Author):

The authors have comprehensively revised the manuscript according to the comments. The concerns have been addressed with additional experimental details. I think that the current version of manuscript can be accepted.